Evolutionary history of a Scottish harbour seal population

Nikolic Natacha natachanikolic@hotmail.com 1 2
Thompson Paul 3
de Bruyn Mark 4
Macé Matthias 5
Chevalet Claude 2
1 ARBRE (Reunion Island Biodiversity Research Agency) , Saint-Leu , La Réunion
2 Génétique Physiologie et Systèmes d’Elevage - UMR1388, INRAE , Castanet Tolosan , France
3 Lighthouse Field Station, Sciences School of Biological Sciences, University of Aberdeen , Cromarty , United Kingdom
4 School of Life and Environmental Sciences, University of Sydney , Sydney , Australia
5 Laboratoire d’Anthropologie Moléculaire et d’Imagerie de Synthèse - UMR 5288, CNRS , Toulouse , France
King Elizabeth
Electronic publication date: 2020 Jul 10
Publication date: 2020
Volume: 8
Electronic Location ID: e9167
Received 2020 Jan 15; Accepted 2020 Apr 19
Copyright: ©2020 Nikolic et al.
Copyright year: 2020
Copyright holder: Nikolic et al.
License: This is an open access article distributed under the terms of the Creative Commons Attribution License, which permits unrestricted use, distribution, reproduction and adaptation in any medium and for any purpose provided that it is properly attributed. For attribution, the original author(s), title, publication source (PeerJ) and either DOI or URL of the article must be cited.
License URL: https://creativecommons.org/licenses/by/4.0/

Keywords: Evolution, Genetic, Seal

Funding: INRAE (FRANCE) Genotoul platform (FRANCE), University of Aberdeen This work was supported by INRAE (FRANCE), Genotoul platform (FRANCE), and University of Aberdeen. The funders had no role in study design, data collection and analysis, decision to publish, or preparation of the manuscript.

==============================
Efforts to conserve marine mammals are often constrained by uncertainty over their population history. Here, we examine the evolutionary history of a harbour seal (Phoca vitulina) population in the Moray Firth, northeast Scotland using genetic tools and microsatellite markers to explore population change. Previous fine-scale analysis of UK harbour seal populations revealed three clusters in the UK, with a northeastern cluster that included our Moray Firth study population. Our analysis revealed that the Moray Firth cluster is an independent genetic group, with similar levels of genetic diversity across each of the localities sampled. These samples were used to assess historic abundance and demographic events in the Moray Firth population. Estimates of current genetic diversity and effective population size were low, but the results indicated that this population has remained at broadly similar levels following the population bottleneck that occurred after post-glacial recolonization of the area.

Introduction

Efforts to conserve marine animals are frequently constrained by uncertainty over historic baselines and the factors driving changes in abundance (Lotze & Worm, 2009). A variety of techniques have been developed to address this issue, including archaeological investigations (Rick & Lockwood, 2013), studies based upon historical records or traditional knowledge (McClenachan, Ferretti & Baum, 2012), and molecular analyses of changes in genetic diversity (Roman & Palumbi, 2003;ref-101).

Harbour seals (Phoca vitulina) are widely distributed around the North Atlantic and North Pacific coasts, but studies over recent decades have identified wide variations in the status of these populations. Abundance in many parts of Europe was severely reduced by successive outbreaks of Phocine Distemper Virus (PDV), but mortality in some populations was low (Heide-Jørgensen, Härkönen & Aberg, 1992; Härkönen et al., 2006). Similarly, longer-term trends in harbour seal abundance show steady increases in some parts of their range (Jeffries et al., 2003; Aarts et al., 2019), whereas other regions have experienced unexplained declines (Boveng et al., 2003; Lonergan et al., 2007; Hanson et al., 2015; Thompson et al., 2019).

These changes have resulted in new conservation measures in many areas, the most significant of which in European waters is the EU Habitats Directive (Baxter, 2001). This requires EU Member States to develop a NATURA 2000 network of Special Areas of Conservation (SACs) for a range of key species, including harbour seals (Thompson et al., 2019). In UK waters, SACs have been established within eleven management units that reflect the current understanding of genetic structure (Olsen et al., 2017), but there is uncertainty over the causes underlying different regional trends in abundance and historic baselines (Matthiopoulos et al., 2014; Thompson et al., 2019). Here, we focus on the evolutionary history of a population that uses one of these SACs, in the Moray Firth, NE Scotland. In this area, individual-based studies since 2006 have provided detailed estimates of contemporary vital rates (Cordes & Thompson, 2014; Matthiopoulos et al., 2014) during a period where there has been no clear trend in abundance (Thompson et al., 2019). The most recent counts indicate that the population has declined by around 40% since the mid 1990’s (Thompson et al., 2019), but direct information on the historic abundance and evolution over multiple generations is lacking given the lack of comparable survey data prior to this. An alternative approach is to assess historic changes in the genetic-based indicator, effective population size (Ne). The effective size reflects the abundance and evolutionary history of the population, and can inform conservation efforts because it affects the degree to which a population can respond to selection (Berthier et al., 2002).

Analysis of neutral molecular markers such as microsatellites can be used to calculate Ne and provide an evolutionary perspective to these conservation and management issues (King et al., 2001). Genetic studies conducted on harbour seals have generally used less than 15 microsatellite markers for “Atlantic” harbour seals (12 in Olsen et al. (2017); 15 in Anderse et al. (2011); 7 in Goodman (1997); Goodman (1998)) and less than 20 for “Pacific” harbour seals (5 in Burg et al. (1999), 8 in Curtis, Stewart & Karl (2011), 7 in Dishman (2011), 20 in Hayes et al. (2006) and 6 in Herreman et al., 2009). We used the markers developed specifically for harbour seal microsatellites (Allen et al., 1995; Coltman, Bowen & Wright, 1996; Goodman, 1997) and a mix of pinniped microsatellite markers (Allen et al., 1995; Buchanan et al., 1998; Davis et al., 2002; Gelatt et al., 2001), as other harbour seal microsatellite papers (ex. Olsen et al., 2017). We therefore tested 30 markers, which allowed us to utilise 17 polymorphic microsatellite markers (25 markers minus 8 monomorphic markers) for genetic analysis of harbour seals. Previous study of genetic structuring of UK harbour seals suggests that there are two main initial groups consisting of localities in the northern UK, and the southern UK and mainland Europe (Olsen et al., 2017). However, Olsen et al. (2017) further divided these groups into geographically distinct genetic clusters, including the North-East England and Eastern Scotland cluster (cf. Fig. 2 of Olsen et al., 2017) that includes our Moray Firth study sites (cf: Fig. 1 of Hanson et al., 2015). Here, we extended the number of microsatellite markers available for harbour seals in the Moray Firth to estimate historic changes in effective population size in this region.

Figure 1 Mean geographic location of the three areas sampled (Dornoch –13 individuals, Cromarty –12 individuals and Inverness –68 individuals) from Moray Firth in north-east Scotland.

Using this extended set of microsatellite markers, the objectives of the present study were to address the following questions: (i) what does current genetic diversity tell us about historic changes of harbour seal effective population sizes; and (ii) how might genetic data contribute to understanding the decline of seals in the north-east of Scotland?

Material & Methods

Study area

The Moray Firth study population (Fig. 1) contained an estimated 1,653 harbour seals in 1993 (Thompson et al., 1997a) when the samples used in this study were collected. Seals come ashore at inter-tidal sites throughout the year, with most pups born in three sub-areas spaced approximately 50 km apart (Dornoch Firth, Cromarty Firth, and Beauly Firth). One of these sub-areas, in the Dornoch Firth, has been designated as a SAC to protect harbour seal populations under the EU Habitats Directive (Cordes et al., 2011; Thompson et al., 2019).

Samples and DNA extraction

Archived blood samples were collected from 93 harbour seals that had been captured, sampled and released as part of earlier ecological studies (Thompson et al., 1997b; Hall et al., 2019). Samples were collected between 1992 and 1995 (see Thompson et al., 1997b for details of capture and sampling methods), and constituted 47 females and 46 males (48 juveniles, 37 adults, 8 sub-adults, data are available at https://doi.org/10.15454/AOZ7JI); representing approximately 6% of the population. All capture and handling methods were carried out in accordance with the approved guidelines and conducted under licences from the UK Home Office. Blood samples were collected under Home Office licence issued to the University of Aberdeen under the Animal (Scientific Procedures) Act 1986 (PPL number 60/01351). These samples were preserved at −20 °C and hydrated with PBS (Phosphate Buffered Saline) before DNA extraction. Genomic DNA was extracted by QIAamp DNA Blood Mini Kit (QIAGEN) (see the Text S1 for more details).

Marker selection and genotyping

Because the number of available harbour seal microsatellite markers was low, we considered all potential microsatellites for this species and additionally from other pinnipeds—Halichoerus grypus, Hydrurga leptonyx (two species belonging to the family Phocidae) and Odobenus rosmarus rosmarus (Odobenidae). A total of 30 markers were tested Appendix S1A. Previously some primers were designed using Primer 3 (Rozen & Skaletsky, 2000) because only the amplified sequences were provided by the authors (ie the primers H12, HL20, HL16, HL15) Appendix S1B. The primers PVC63 were redefined using Primer 3 to optimize amplification. We retained 25 loci that amplified reliably (meaning that on 12 individuals tested a minimum of 10 were successfully amplified and visualized on agarose gel) for harbour seals Appendix S1B, with a minimum of twelve repeat motifs and a GC percentage of approximately 50%.

All 93 harbour seal samples were successfully genotyped with these 25 microsatellite markers using fluorescent labelled primers and multiplex PCR pools (Appendices S1A, S1B). PCR amplification was carried out using an Applied Biosystems 2720 Thermal Cycler with 10 µl reaction volume containing ∼50 ng DNA, 1.5 mM MgCl2, 1x Promega buffer, 200 μM dNTPs, 0.5 µM each primer, 0.5 U Taq DNA polymerase (Promega). An initial denaturation step at 94 °C for 5 min was followed by 42–45 cycles of 30 s at 94  °C, 30 s at Tm °C (annealing temperature), 30 s at 72 °C, followed by a final elongation step of 30 min at 72 °C. The annealing temperatures were optimised for each locus Appendix S1B. The PCR products (2 µl) were added to a mixture of deionised formamide and the internal size standard GENESCAN-400HD Rox (Applied Biosystems) (8 µl), then denatured for 5 min at 95 °C. For this mixture (formamide and internal size standard), we prepared a final volume of 1000 µl with 982.5 µl of formamide and 17.5 µl GENESCAN-400HD Rox. Individual electropherograms were obtained using an ABI 3730 multi-capillary sequencer. PCR products were visualised by GeneMapper v4.0 software (Applied Biosystems) (see the data and profile examples per pool at https://doi.org/10.15454/AOZ7JI). Across the panel of 25 markers, 8 were monomorphic Appendix S1B in both male and female individuals. The genetic analysis was performed on the overall panel (25 markers) and on the polymorphic markers (17 markers).

Analysis of diversity and genetic differentiation

To ensure that the number of loci was sufficient, we calculated the probability of individual identity (PI; the probability that two individuals in a population have identical genotypes) for each locus and their combinations with the program GENALEX (Peakall & Smouse, 2006; Peakall & Smouse, 2012). SPOTG (Hoban, Gaggiotti & Bertorelle, 2013) was used to estimate the power of individual assignment, using 1,000 runs, the number of sampled individuals (93), the mean number of alleles (4), and the FST values found in this study under a normal allele frequencies model.

Genetic diversity was measured as the mean number of alleles per locus (A) for 25 markers. Excluding the monomorphic markers we measured observed heterozygosity (Ho), expected heterozygosity (He), and Nei’s (1978) unbiased heterozygosity (H.n.b) using GENETIX 4.05.2 (Belkhir et al., 1998). Estimates of homozygote and heterozygote excess that differed significantly from zero (p < 0.05) were calculated from the standard error rates with confidence intervals in Pedant (Johnson & Haydon, 2007). Deviations from Hardy-Weinberg Equilibrium (HWE) were assessed using polymorphic loci, and exact tests with p-values and their standard errors were computed using ARLEQUIN version 3.1 (Excoffier, Laval & Schneider, 2005; Excoffier & Lischer, 2010) with permutations (1,000,000 chains and 100,000 steps). Polymorphism information content (PIC) was generated in Cervus (Kalinowski, Taper & Marshall, 2007). Probability of parentage exclusion (PE1, single parent (Jamieson & Taylor, 1997); PE2, a second parent given a first parent assigned (Jamieson, 1994); PE3, a pair of parents (Jamieson & Taylor, 1997)) was estimated per locus using INEst (Chybicki & Burczyk, 2009). The potential occurrence of null alleles and scoring errors due to stuttering or large allele dropout in the data set was assessed using the software MICRO-CHECKER (Oosterhout et al., 2004), and the significance of null allele frequency (Fnull) was estimated with INEst using the individual inbreeding model with 100,000 iterations (estimates significantly different from zero, p < 0.05). The inbreeding coefficient FIS was estimated from polymorphic loci with 10,000 bootstraps by GENETIX software. FIS measures the decrease in heterozygosity due to inbreeding, assortative mating, or selection. Genotyping error rate per allele, E1 referring to allelic dropout rate and E2 to the false allele rate, and the 95% confidence interval (CI) with 10,000 permutations, were evaluated with maximum likehihood from 10 individuals’ random replicates by marker based on He computed in Pedant. The number of repeated genotypes (Nrep) and the percentage (%) of the total number of individuals genotyped for each loci were also estimated.

Our findings were consistent with the previous study identifying Moray Firth sites as part of one genetic cluster (Olsen et al., 2017). Thus, we also investigated whether external migrations (e.g., gene flow from another cluster) were possible because grouping several different populations could skew our analyses of evolutionary history. Genetic differentiation was visualized by Factorial Correspondence Analysis (FCA) in GENETIX using the polymorphic loci (n = 17). Pairwise Wright’s F-statistics FST (Weir & Cockerham, 1984) and their levels of significance were assessed based on 10,000 permutations using ARLEQUIN 3.1 (Excoffier, Laval & Schneider, 2005), considering geographical location per individual. Number of migrants (Nm) was estimated through the frequency of private alleles with GENEPOP (Raymond & Rousset, 1995; Rousset, 2008) online (http://genepop.curtin.edu.au). We estimated only the level of migration rates (m) (superior or inferior to 0.1, and not the values of gene flow due to uncertainty with a weak genetic structure) using the Bayesian multi-locus genotyping approach implemented in BAYESASS 3.0.4 (Wilson & Rannala, 2003) with different random number seed values (S) and sampling frequency (n), 10,000,000 iterations, and a burn-in period of 1,000,000. To detect immigration into the Moray Firth, we used Rannala & Mountain’s (1997) estimate and ran Monte-Carlo simulations with 10,000 repetitions implemented in GENECLASS2 software (Piry et al., 2004), considering geographical location per individual.

Finally, we used a population assignment test to check for sex-biased dispersal using the software GENALEX. This method produces an Assignment Index correction (AIc) for each sex following the method of Mossman & Waser (1999). Negative AIc values characterize individuals with a higher probability of being migrants (high dispersion) and positive values characterize individuals with a lower probability of being migrants. Mean AIc values were compared for each sex with a non-parametric Mann–Whitney U-test using R version 3.0.2 software. Moreover, hierarchical Analysis of Molecular Variance (AMOVA) analyses were performed, with 9,999 permutations, for each sex on their own, and for the combined male and female data set to determine whether genetic variation was similar for males and females.

Clustering analysis

STRUCTURE 2.3 (Pritchard, Stephens & Donnelly, 2000) was used to identify population structure and individual admixture coefficients. Five independent runs were performed on STRUCTURE for each assumed number of population(s) K = 1–6 under an admixture model. All runs were executed with 50,000 burn-in periods and 200,000 MCMC (Markov chain Monte Carlo) repetitions, using the three regions defined earlier as prior information. STRUCTURE HARVESTER 0.6.94 (Earl Dent & VonHoldt Bridgett, 2012) was used to visualise results and assess K, the number of genetic populations that best fit the data, based on Maximum Likelihood (Evanno & Goudet, 2005). Structure Selector (http://lmme.qdio.ac.cn/StructureSelector/) was also used to process the Puechmaille (2016) approach on uneven sampling and to also estimate the best K. Plots for optimal K was performed with CLUMPAK (Kopelman et al., 2015).

In addition, a discriminant analysis of principal components (DAPC, Jombart, Devillard & Balloux, 2010) was performed using R package ADEGENET (Jombart, 2008; Jombart & Ahmed, 2011). DAPC is a multivariate analysis that integrates principal component analysis (PCA) with discriminant analysis to summarize genetic differentiation between groups (Jombart, 2008). Sampling location was used as prior. While STRUCTURE forms genetic clusters of individuals by minimizing departure from Hardy–Weinberg and linkage disequilibria, DAPC maximizes genetic separation among groups and minimizes variation within groups (Jombart, Devillard & Balloux, 2010) which may constitute a more accurate approach for species exhibiting potentially high gene flow (Bailleul et al., 2018).

Isolation by distance was tested using a Mantel test between genetic (Euclidean Edwards’ distance) and geographic distances with 10,000 resamplings between individuals and regions using ADEGENET, ADE4 (Thioulouse et al., 1997) and GRDEVICES (R Development Core Team and contributors worldwide) R packages.

Effective population sizes and evolutionary history

The average mutation rate on the set of loci (µ), ancestral time (Tf), and present and ancestral effective size (No and Na) were estimated for the seals using MSVAR (Beaumont, 1999), running 80,000 MCMC chains and 20,000 iterations between chains. This method is suited to microsatellite data that are assumed to be evolving by a stepwise mutation model (SMM), sampled from a population that has varied in size. When compared with other classic coalescence models (TM3 and DIYABC) aimed at estimating the present and ancestral effective sizes, MSVAR was most efficient for small populations (Nikolic et al., 2009). Because we used this model only to get preliminary results for the seal mutation rate, and computation times are very long (here more than 3 days), this analysis was run only with the polymorphic panel (17 markers). MSVAR estimates the effective population size for the present and ancestral effective sizes but does not infer fluctuation of effective size between them. We applied Gelman & Rubin, 1992’s (1992) test to monitor convergence of MCMC output.

To check for historic population declines in the seals, we used BOTTLENECK software (Piry, Luikart & Cornuet, 1999) with 10,000 iterations and we applied three tests—the sign test, the standardized differences test (Cornuet & Luikart, 1997), and the Wilcoxon sign-rank test (Luikart et al., 1997) to analyze the presence of heterozygote excess resulting from perturbation of allele frequencies. We applied the Wilcoxon signed-rank test specifically as it does not require a large number of polymorphic loci which are scarce in a population with low variability (Han et al., 2010). Inferences from heterozygosity excess or deficiency tests are heavily influenced by the mutational model (Busch, Waser & DeWoody, 2007). Here, we used the Stepwise Mutation model as it is thought to be a more appropriate model for use with microsatellites (Nikolic & Chevalet, 2014b).

Bayesian methods using coalescence theory and MCMC sampling to estimate posterior distributions of demographic parameters and history, seem more robust to certain violations of mutation model assumptions (Girod et al., 2011) and bottleneck duration (Peery et al., 2012). Hence, to estimate historic fluctuations of effective population size from microsatellite markers, we used the algorithm and method VarEff (Chevalet & Nikolic, 2010; Nikolic & Chevalet, 2014b) programming in R package. The model assumes a stepwise mutation model for microsatellites and makes use of an approximate likelihood of data based on theoretical results. It is then implemented in a MCMC framework which simulates past demography by sampling step functions. The model is freely available as an R package (https://forge-dga.jouy.inra.fr/projects/package-vareff-variation-of-effective-population-size/). Results were based on MCMC chains including 10,000 dememorization steps, a total length of 1,000,000 and the extraction of 10,000 uncorrelated states as suggested by previous analysis (Nikolic & Chevalet, 2014b). Results are provided in normalized scales θ ˆ = 4N ˆ µ  for population size (4* effective size*mutation rate) and T = gµ for time (number of generations *mutation rate), or in the natural scale, effective size Ne and generation number g provided the mutation rate µis known. For harbour seals, we set µ= 0.00015 and used the generation time of 8.75 years based upon the value used in Swart, Reijnders & Van Delden (1996).

VarEff offers several functions to characterize the posterior distribution of effective size at several times in the past: arithmetic and harmonic means, median and mode of the distribution, detailed distribution at specified times, with the quantiles and standard deviation. In addition, the results of VarEff allow the posterior distribution of the Time to the Most Recent Common Ancestor (TMRCA) of two random alleles to be recovered, which provides complementary information on the occurrence and time of past bottlenecks, since peaks in this distribution indicate times when coalescence events likely occurred, and intervals between peaks may indicate periods when bottlenecks occurred. Depending on the intensity and shape of the peak and the value of effective size (obtained with the previous functions), information on the bottleneck event(s) can be recovered. Futhermore, because migration can mimic the effects of bottlenecks (Nikolic & Chevalet, 2014b), we repeated the same analysis discarding the immigrants that were detected (2 individuals).

Results

Genetic polymorphism and panel of markers

Of the 25 microsatellite loci, eight loci were monomorphic for both sexes (PVC63, PVC74, SGPV3, PVC26, PVC29, GS1, Hl-16, and OrrFCB23) and one marker Hl-20 was monomorphic in males Appendix S1B. These markers were hence excluded from the analyses resulting in a final panel of 17 polymorphic markers (Table 1). Our final genotyping data contained only 3.2% missing data for 93 individuals and 17 polymorphic markers.

Table 1 Summary statistics of the 17 microsatellite markers selected for Harbour seals (Phoca vitulina).

Sample sizes per locus (S). Number of alleles (A). Expected (He), unbiased Nei’s95 expected (H.n.b) and observed (HO) heterozygosity. Hardy–Weinberg equilibrium (HWE) p-values (P) with the standard error in parentheses. Polymorphism information content (PIC). Probability of identity (PI). Probability of parentage exclusion (PE1, single parent; PE2, a second parent given a first parent assigned; PE3, a pair of parents). Null allele frequency (Fnull). Number of repeated genotypes (Nrep and percentage (%) of the total number of individuals genotyped for each loci). Genotyping error rate per allele, E1 referring to allelic dropout rate and E2 to the false allele rate, and the 95% confidence interval (CI). Significant values are highlighted in bold (P < 0.05) for heterozygote excess.

														Genotyping error rate	
Locus	S	A	He	H.n.b	Ho	P	PIC	PI	PE1	PE2	PE3	Fnull	Nrep(%)	E1 (CI 95%)	E2 (CI 95%)	
SGPv9	85	3	0.277	0.279	0.329	0.254 (0.00038)	0.242	0.558	0.038	0.123	0.205	0.023	10 (12)	0.00 (0.00–0.41)	0.00 (0.00–0.07)	
SGPV11	92	3	0.275	0.276	0.174	0.002 (0.00004)	0.240	0.561	0.038	0.122	0.203	0.107	10 (11)	0.00 (−0.00–0.78)	0.00 (0.00–0.07)	
SGPV17	92	3	0.409	0.411	0.413	0.019 (0.00013)	0.345	0.413	0.084	0.184	0.292	0.039	10 (11)	0.00 (−0.00–0.20)	0.00 (−0.00–0.07)	
SGPV10	90	2	0.231	0.232	0.267	0.352 (0.00047)	0.204	0.618	0.027	0.102	0.174	0.026	10 (11)	0.00 (0.00–0.40)	0.00 (−0.00–0.07)	
SGPV16	74	14	0.886	0.892	0.919	0.659 (0.00033)	0.876	0.023	0.628	0.772	0.921	0.013	10 (14)	0.00 (0.00–0.08)	0.00 (−0.00–0.07)	
PVC19	93	2	0.350	0.352	0.344	1.000 (0.00000)	0.289	0.484	0.061	0.144	0.229	0.039	10 (11)	0.00 (−0.00–0.19)	0.00 (0.00–0.07)	
PVC30	82	4	0.493	0.496	0.476	0.767 (0.00037)	0.388	0.362	0.122	0.204	0.310	0.039	10 (12)	0.00 (−0.00–0.21)	0.00 (0.00–0.07)	
PVC78	93	4	0.243	0.245	0.204	0.003 (0.00005)	0.219	0.597	0.030	0.113	0.193	0.058	10 (11)	0.00 (−0.00–0.19)	0.00 (0.00–0.07)	
GS7	93	2	0.157	0.158	0.172	1.000 (0.00000)	0.145	0.723	0.012	0.072	0.129	0.033	10 (11)	0.00 (−0.00–0.71)	0.00 (0.00–0.07)	
GS2	91	3	0.173	0.174	0.165	0.557 (0.00049)	0.163	0.694	0.015	0.086	0.156	0.047	10 (11)	0.00 (−0.00–0.25)	0.00 (0.00–0.07)	
GS3	93	4	0.560	0.563	0.570	0.579 (0.00048)	0.491	0.262	0.158	0.294	0.442	0.022	10 (11)	0.00 (−0.00–0.16)	0.00 (0.00–0.07)	
H12	93	5	0.614	0.617	0.613	0.901 (0.00025)	0.550	0.213	0.194	0.345	0.504	0.029	10 (11)	0.00 (−0.00–0.09)	0.00 (0.00–0.07)	
HL20	90	2	0.043	0.044	0.044	1.000 (0.00000)	0.043	0.916	0.001	0.021	0.041	0.050	9 (10)	0.70 (−0.03-1.81)	0.00 (−0.00–0.14)	
HL15	93	3	0.242	0.243	0.237	0.707 (0.00043)	0.215	0.601	0.029	0.109	0.186	0.042	10 (11)	0.00 (−0.00–0.76)	0.00 (0.00–0.07)	
OrrFCB2	90	4	0.547	0.550	0.633	0.070 (0.00022)	0.489	0.263	0.151	0.296	0.449	0.016	10 (11)	0.00 (−0.00–0.11)	0.00 (−0.00–0.07)	
OrrFCB1	79	2	0.013	0.013	0.013	1.000 (0.00000)	0.012	0.927	0.001	0.019	0.037	0.055	10 (13)	0.70 (−0.03-1.81)	0.00 (0.00–0.12)	
OrrFCB24	90	3	0.510	0.513	0.956	0.000 (0.00000)	0.391	0.359	0.130	0.202	0.303	0.009	9 (10)	0.00 (0.00–0.08)	0.00 (−0.00–0.08)	
Mean	89	3.7	0.354	0.356	0.384	0.522 (0.000)	0.312									

The probability of identity (PI) values ranged from 0.023 to 0.927 and the probability of exclusion (E1, E2, E3) from 0.001 to 0.921 (Table 1). A total of 9 markers had a probability of identity in the higher range at 0.5 (Table 1). The analysis based on the cumulated probability of individual identity (PI), an indication of the statistical power of marker loci, revealed that 10 to 17 polymorphic markers were sufficient to carry out the population genetics analysis of harbour seals in the Moray Firth (Fig. 2). SPOTG estimated that 93 individuals and 17 microsatellite markers could detect power of individual assignment >57% and connectivity >87%. However, the theoretical study by Nikolic & Chevalet (2014b) shows that the number of markers to assess accurate effective population size must be higher than the basic diversity analyses. It was therefore important to have more than 10 markers for this study to reduce the variance of the estimators around the effective size values.

The total number of alleles for the remaining 17 loci ranged between 2 and 14 (Table 1), while the mean number of alleles was 3.7 (Table 1). The global mean of observed heterozygosity was 0.384 for the 17 polymorphic microsatellites. Estimates of homozygote and heterozygote excess were not significant except OrrFCB24 with a heterozygote excess (Table 1).

Genotyping error rates and associated 95% confidence intervals were very close to zero for all loci except HL20 and OrrFCB1 (Table 1). Most loci were close to HW equilibrium (P > 0.05, Bonferroni correction applied) (Table 1) except SGPV11, SGPV17, PVC78 and OrrFCB24 (Table 1). The locus SGPV17 has been suspected to be X-linked in pinniped species (Coltman, Bowen & Wright, 1996; Gemmell et al., 1997; Pastor et al., 2004), although this remains equivocal (Herreman, Blundell & Ben-David, 2008). Such a linkage would imply that males are homozygous for this marker. In the present study, we identified 3 alleles (153, 155 and 161) at SGPV17 with similar allelic frequencies in males and females. Heterozygote frequencies were 14/46 in males and 24/47 in females suggesting no X-linkage for this locus. Analysis of sex-biased dispersal using the marker SGPV17 showed there were no significant differences in the mean AIc with the polymorphic panel (Mann–Whitney U-test, p-value = 0.78). This result indicates no sex-biased dispersal. Thus, we concluded that we have no reason to exclude the marker SGPV17 in the polymorphic panel on the basis of linkage.

Figure 2 Probability of Identity for each Locus (PI) and for Increasing Combinations (PIsibs) with the 17 polymorphic microsatellite markers genotyped on harbour seals in the Moray Firth.

MICROCHECKER estimated that only SGPV11 showed evidence for a null allele but the test by permutations with INest revealed that the result was not significant (Table 1). Null allele analyses were not significant for all loci (Table 1). Hence, there was no evidence for scoring error due to stuttering and no evidence of large allele dropout. The harbour seal population appears to be in Hardy Weinberg equilibrium with the global test (P = 0.519).

With respect to the distribution of repeat numbers Appendix S1C, harbour seals in the Moray Firth exhibited quite large distances between microsatellite alleles (up to 37 repeat numbers) and a large range of missing values (between 10 and 24, and between 26 and 36) that may be the signature of bottleneck events.

Genetic differentiation and clustering analysis

Analysis of sex-biased dispersal (Mossman & Waser, 1999) showed no significant differences in the mean AIc with the polymorphic panel (Mann–Whitney U-test, p-value = 0.69), indicating no sex-biased dispersal. Futhermore, when males and females were considered separately in AMOVA analyses, no genetic differentiation was found between the sexes for the panel of polymorphic markers.

FIS values by permutations (sampling with replacement of individuals with all loci) were negative −0.081 (with CI narrow and negative, −0.129 and −0.044) and by Jacknife result on locus −0.079 (variance = 0.006). Negative FIS values indicate that there were more heterozygotes than expected and individuals in the population may be less related than expected under a model of random mating. Global frequency distribution of observed and permuted FST values were not significant, as they lie well inside the distribution of FST for the null hypothesis. FST were <0.05 between the three geographic areas with only the value between Dornoch and Inverness (0.014) being significant according to permutation tests Appendix S1D. Corrected average pairwise differences were low and not significant Appendix S1D. These values indicate an absence of significant differentiation. Clustering analyses (STRUCTURE and DAPC) supported the existence of one main cluster with no spatial differentiation and an absence of structuring between sampling localities Appendix S1E and sex. Moreover, the isolation by distance (Mantel test) was clearly not significant (p-value = 0.12).

Concerning migration, FCA on individuals identified 4 axes with eigenvalues of 0.15 (axis 1), 0.089 (axis 2), 0.078 (axis 3), and 0.072 (axis 4). The three dimensional FCA Appendix S1F revealed two migrants as immigrants (one female from Cromarty and one male from Inverness), i.e., a rate of 2.15%. The mean Nm value using private alleles, after correction for size, and considering the individual geographic localisation was Nm = 4.52. According to Wright (1969), Nm <1 indicates strong genetic differentiation, and Nm much larger than 1 means that panmixia can be assumed for the localities of the population. Based on Hastings (1993), Faubet, Waples & Gaggiotti (2007) suggest that m <0.1 is needed to ensure demographic independence of populations. In this study, m were higher than 0.1 between sampling localisations in the Moray Firth.

Thus, the genetic differentiation and clustering analysis indicate that the harbour seal populations in the Moray Firth behave as a single genetic group. When considering the Moray Firth as a single genetic group, mean frequency of private alleles was equal to 0.03 which is consistent with the two immigrants detected by FCA. The Bayesian method of Rannala & Mountain (1997) did not detect immigrants into the Moray Firth. However, based on frequencies of Paetkau et al. (1995) and 10,000 resamplings, 5 of the 93 individuals were detected as immigrants (4 males and one female: 5%). Nei’s standard distance (1972) with 10,000 resamplings with the algorithm of Paetkau et al. (2004) showed 4 immigrant individuals (2 females and 2 males), equivalent to around 4%. Hence, the global immigration rate in the Moray Firth is in the range of 2 to 4%.

Evolutionary trends

Most markers shared a similar range of mutation rates of around 0.0001–0.0002 with MSVAR estimates. VarEff provided global estimates of θ ˆ = 4 N ˆu, roughly corresponding to present (θ ˆo = 0.77), ancestral (θ ˆa = 38.90), and intermediate times (θ ˆI = 8.56) that help fix priors for effective size.

The global estimates in the present effective sizes of VarEff provided results in the same order of magnitude as those from MSVAR for seals particularly in terms of median but with lower standard deviations (Sd) (Table 2).

Table 2 Mean (arithmetic), median and standard deviation of present (No) and ancestral (Na) effective population size, the mutation rate (µ), and ancestral time in years (Tf) of Harbour seals in Moray Firth (left part, MSVAR analysis).

Mean (arithmetic and harmonic), median and standard deviation (Sd) of present (No) and the estimated size 5.000 generations ago (N5000), and the global present (N ˆo) and ancestral estimate (N ˆa) (right part, VarEff analysis).

	MSVAR	VarEff	
	No	Na	µ	Tf	No	N5000	N ˆo	N ˆa	
Mean	1,669	342,874	0.00018	96,416	988 and 649	14,824			
Median	821	128,186	0.00010	37,235	714	11,298	1,287	64,826	
Sd	2,979	846,836	0.00026	219,518	2,089	230,55			

According to Gelman and Rubin’s test (Gelman & Rubin, 1992), convergence in the MSVAR estimations of the effective size occurred after 20,000 iterations; and run 4 was the best supported Appendix S1G. The results from MSVAR suggested that the effective size of the Moray Firth harbour seal population during our study period (1990–1995) was 821–1,669 (No) (sd 2,979), and that the ancestral effective size was much higher around 128,000–340,000 (Na) (sd 846,836) (Table 2). Values for mutation rates were around 0.0001–0.0002 (µ) (sd 0.00026), and the ancestral coalescent time was estimated to occur 37,000–96,000 years ago (Tf) (sd 219,518) (Table 2).

VarEff harbour seal effective size priors were set between N ˆo and N ˆi from the global current (θ ˆo) and intermediate (θ ˆi) theta (4 N ˆu) estimates: the prior mean was set to N ˆ = 5,000 with a large variance (equal to 3 as suggested by Nikolic & Chevalet (2014a) for the logarithms of N). The estimate of present effective size (mean harmonic (No) and N ˆo) was around 700 and 1,300 and the ancestral effective size (N ˆa) 65,000 (Table 2). We also analysed females and males separately and obtained similar effective sizes which is consistent with no sex-biased dispersal. The arithmetic mean, harmonic mean, mode, and median of the posterior distributions for effective population sizes of harbour seals revealed a trend pattern that looks similar with a recent decrease from a higher effective size (Fig. 3A). A decrease in effective size of around 2.5% was suggested during the last 25 generations. The effective size at generation 0 (No) was 649 for the harmonic mean, 988 for the arithmetic mean, 714 for the median and 729 for the mode. This represents a 388–2,181 95% confidence interval. The difference observed between estimators of past Ne (mean, median and mode) (Fig. 3A) is likely to be due to the long tail of the posterior probability distribution (Nikolic & Chevalet, 2014a).

Figure 3 Effective size (Ne) of harbour seals in the north-east UK (from Moray Firth genetic group) as a function of past generation time (G) using 17 microsatellites (VarEff analysis).

(A) Arithmetic (red) and harmonic (green) mean, mode (blue), and median (black) from sampling time (0) to 30,000 generations ago. (B) Posterior densities at the past generation time (G): 2,000 (black), 3,000 (blue), 4,000 (red), 5,000 (green), 6,000 (grey), 7,000 (purple), 8,000 (orange), 9,000 (pink), and 10,000 (red) generations ago. (C) Posterior densities at the past generation time (G) 10,000 (black), 20,000 (blue), 30,000 (red), 40,000 (blue), 50,000 (grey), 60,000 (purple), 70,000 (orange), 80,000 (green), 90,000 (brown), and 100,000 (red) generations ago.

According to global estimates (θ ˆo, θ ˆi, θ ˆa) and effective size distribution (Fig. 3) from VarEff, harbour seals in the Moray Firth came from a large ancestral population. The estimated effective sizes in the past suggest that the extant population derived from an ancestral population of around 10,000–15,000 individuals approximately 5,000-30,000 generations ago (Fig. 3). This seems to be preceded by a lower ancient size (around 2,000–3,000 individuals) some 30,000–100,000 generations ago (Fig. 3C); however this should be considered with caution, given the flat posterior distribution in ancient times. Concerning the recent and historical effective size, several decreases of the effective size were observed. Firstly, a long period of decline over the last 2,000 generations (17,500 years) was suggested (Figs. 3A, 3B), when population size decreased from tens of thousands to less than 1,000 (Figs. 3A, 4A) and followed by a further decline prior to 600 generations ago (Fig. 4), estimated at around 800 generations ago (Fig. 5). Finally, a recent accelerated decline during the last 100 generations was followed by a period when population size remained approximately constant (Ne between ≈ 665 and 685) (Fig. 4).

Figure 4 Harbour seal’s effective size (harmonic mean of the posterior distribution, VarEff analysis) within Moray Firth (using 17 microsatellites) in generation time (from 0 to 1,000 generations ago) (A) and in calendar years (from 1995 (sampling date) to 500 AD.

The arrows represent the main trends: reduction (red) and increase (blue). (B) shows the latest tendency enlarged, the last red arrow.

Figure 5 Posterior distribution of the Time to Most Recent Common Ancestor allele (TMRCA, VarEff analysis).

Suggested coalescent events are given as generation numbers and as years for harbour seals in the Moray Firth (Scotland). Each peak, in the posterior distribution, represents a potential bottleneck.

The VarEff results describing the global evolution of the population are consistent with the tests of heterozygosity excess (sign, standardized and Wilcoxon tests) assuming SMM as proposed in BOTTLENECK Appendix S1H. The latter analysis suggested a recent bottleneck in the north-eastern UK population. Bayesian methods make use of coalescence theory and MCMC sampling to characterize demographic history and can have a higher probability of detecting bottlenecks than heterozygosity excess tests (contrasts heterozygosity expected under Hardy–Weinberg equilibrium with heterozygosity expected under mutation-drift equilibrium calculated from observed number of alleles) (Cornuet & Luikart, 1997; Girod et al., 2011; Peery et al., 2012). Concerning the coalescent results from VarEff, Fig. 4A represents the harmonic mean of the last 1,000 generations (A), which was characterized by a decrease followed by a slight increase before a further decrease in the last 100 generations. Variation over the last 1,500 years is also shown in Fig. 4B, based upon the assumed generation time of 8.75 years.

According to Fig. 5, which shows the distributions of TMRCA, the harbour seal population underwent drastic bottlenecks. The more recent times of the bottlenecks were estimated in generation time (G) and in years, assuming a generation time of 8.75 years. Figure 5 also reveals that huge coalescence events occurred between 7,000–10,000 years (around the first peak in Fig. 5). These coalescence events occurred after a period of decline of the population size (the drastic bottleneck seen around 17,000 years ago, the time when the density is at a minimum between the two peaks). Before that period, coalescence events occurred at times given by the second peak of the distribution, which corresponds to the large ancestral population size, as illustrated in Figs. 3 and 4. Harbour seal showed declines over the last 1,000 years, with a steeper decline over the last few centuries. Futhermore, the analysis run without the two detected immigrants provided exactly the same results.

Discussion

Genetic diversity

Results from the PI and SPOTG simulations suggest that the number of individuals and polymorphic markers are sufficient to provide high assignment discrimination and to detect evolutionary and ecological processes. The low genetic diversity in Moray Firth harbour seals, with an overall average heterozygosity of 38% and 3.7 alleles per microsatellite locus, was lower than in many pinnipeds (summarised in Curtis, Stewart & Karl, 2011). However, this value falls within the range recorded for other North Atlantic harbour seal populations, which were estimated between 24% (Coltman, Bowen & Wright, 1996), 38% (Olsen et al., 2017) and 50% (Goodman, 1998). In contrast, higher levels of heterozygosity (>65%) have been recorded in the North Pacific (Burg et al., 1999). There is concern that small populations with low genetic diversity may become susceptible to environmental, demographic, and genetic stochasticity, increasing their risks for extinction (Franklin, 1980; Newman & Pilson, 1997; Brook et al., 2002). However, low genetic diversity appears to be common amongst many marine mammals (Hoelzel, Goldsworthy & Fleischer, 2002), and the conservation implications of contemporary patterns are difficult to assess without some understanding of the relative role of natural and anthropogenic influences on historic population sizes. For example, low levels of genetic diversity may simply exist in some populations because of the nature of their social structure (Whitehead, 1998), or due to founder effects that occurred following natural post-glacial habitat change (Palsbøll, Heide-Jørgensen & Dietz, 1997). It is likely to be of greater concern to managers where low diversity has resulted from more recent over-exploitation (Hoelzel et al., 1993) or bycatch (Pichler & Baker, 2000).

Genetic structure

Our data suggest that the Moray Firth acts as one genetic group, in accordance with the results of Olsen et al., 2017. Migration per generation above 5–10% can strongly influence the estimate of effective size (Waples & England, 2011) and it was therefore important to exclude this potential bias by evaluating gene flow. Current gene flow of harbour seals into the Moray Firth was low (immigration 2–4%), likely because the adults are strongly philopatric (Thompson & Hall, 1993; Härkönen & Harding, 2001). Goodman’s (1998) study of European harbour seal population differentiation suggests a critical range over which animals are philopatric of around 485 km. In that study, samples collected in the Moray Firth (Scottish east coast) were further than 500 km from neighbouring genetic populations (Scottish west coast, Olsen et al., 2017). In a Neighbor-joining phenogram presented in the Goodman study (1998), harbour seals from the Scottish east coast were distinguable from the Scottish west coast with only 7 microsatellite loci, even though the number of seals from the Scottish west coast was very low (18 individuals). In a recent genetic clustering analysis with 12 microsatellite loci (Olsen et al., 2017), north-west and north-east UK were considered two geographically distinct genetic clusters. The genetic homogeneity between the Moray Firth and other north-eastern localities (Orkney and Shetland; Olsen et al., 2017), and associated low immigration, suggest that we can develop a robust reconstruction of evolutionary history for the north-eastern Scottish population.

Effective population size (Ne)

Inferring Ne from an estimate of θ can be obtained assuming a mutation rate. We estimated the mutation rate at around 0.0001–0.0002, similar to that found in ringed seals (Phoca hispida) (Palo et al., 2003), allowing us to estimate effective population size. Using the complete panel of polymorphic markers with and without immigrants identified by the assignment method, the effective size estimated by the coalescent tool (VarEff) at generation 0 (No) was around 500–1,000 (CI [388–2181]). The advantage of VarEff is that it can estimate the recent historic abundance from theta (θ ˆo) (average of the effective size on the last evolutionary stage) and the current abundance from No (effective size at generation time 0). Ne is a theoretical measure of an idealized population size that would be expected to experience the same rate of genetic diversity loss due to genetic drift (Wright, 1931). Ne is hence defined as the size of an ‘ideal’ population with the observed rate of genetic drift (Wright, 1931; Wright, 1969) and thus does not represent census population size (Palsbøll et al., 2013). In ecology, Ne is usually thought of as the number of breeding individuals that successfully transmit their genes to the next generation (Frankham, 1995) and thus should equal the ‘genetic’ effective population size (Palsbøll et al., 2013). In practice, the exact relationship is rarely known and most studies apply a generic ratio representing a range of estimates (Palsbøll et al., 2013). Small Ne with no or limited gene flow among populations tend to accelerate stochastic loss of genetic diversity and can increase population risk as it leads to inbreeding depression and reduced fertility, and increases the potential fixation of deleterious alleles (Fagan & Holmes, 2006; Gilpin & Soulé, 1986; Palstra & Ruzzante, 2008; Lonsinger, Adams & Waits, 2018). Franklin (1980) suggested a minimum Ne ≥ 50 may be required to avoid short-term inbreeding depression, and an Ne ≥ 500 may be necessary to maintain long-term adaptive potential. However this rule (50/500) is open to criticism as the estimates do not take into account the force of selection. According to Lande (1995), wild populations could not bear the same consanguinity as farm populations (purged by humans); wild populations would fall more quickly into inbreeding depression and therefore the 50/500 rule would be underestimated. Lande (1995) recommended an Ne of 5,000 for long-term viability. Other analyses (Allendorf & Ryman, 2002) suggested an Ne of 1,000 to prevent accumulation of harmfull mutations. All these recommended values must be taken with caution, as they ignore uncertainty arising from environmental and demographic factors, but they encourage conservation of small populations (Grove, 2003).

Ne is typically smaller than census population size (N) and this ratio may help for assessing the genetic health of a population and for predicting short-term and long-term risk (Palstra & Ruzzante, 2008). This ratio varies from 10−5 in many marine invertebrate species to nearly 1.0 in some terrestrial vertebrates (Frankham, 1995; Hedrick, 2005), but comparative data for other pinnipeds are difficult to interpret due to the high level of uncertainty in census population size (Curtis, Stewart & Karl, 2011). Surveys during the sampling period resulted in a direct estimate (N, census population size) of approximately 1,650 individuals (1993) for Moray Firth locality (Thompson et al., 1997a), 773 individuals in 1992 and 575 in 1994 for Firth of Tay and Eden Estuary (Hanson et al., 2015), 9,000 and 6,000 individuals in 1997 for north coast and Orkney, then Shetland respectively (Fig. 4 from Duck & Morris, 2014). These different values suggest that the ratio No/N can be very different depending on sampling location considered. The No estimate in this study represents one genetic cluster, the north-east UK, and including all N would be unwise until we verify that No from other localities are in the same general estimate. Although N is a relatively straightforward entity, it can be difficult to obtain (Palsbøll et al., 2013). Hence, in order to infer N from Ne, a rigorous estimate of the ratio is warranted (Palsbøll et al., 2013).

Evolutionary trends

Over 80% of UK harbour seals are found in Scotland and estimating the historic effective size of harbour seal in the Moray Firth showed that seals have recently experienced a decline, as also revealed by direct counts in parts of both the north-east Atlantic (Thompson et al., 2019; Lonergan et al., 2007; Hanson et al., 2015) and the north-west Atlantic (Bowen et al., 2003). Concerning the ancestral harbour seal population size (MRCA) in the Moray Firth, both models suggested that effective population size had once been extremely large; in the region of 65,000–130,000 derived from ancestral theta for VarEff and the median for MSVAR. One advantage of VarEff is the ability to infer demographic history from a single temporal sample, rather than depending on two or more samples that span multiple generations. Detecting past bottlenecks using the method relies on observing pairs of alleles that have coalesced during the bottleneck event (Nikolic & Chevalet, 2014a). The VarEff method enables various mutation models to be considered, and some trials determined that the risk of false bottleneck detection due to an inappropriate mutation model would be unlikely. Results also indicated that the Moray Firth harbour seal population has undergone a drastic bottleneck, which is concordant with the analysis of detection by the heterozygote deficiency tests. Figures 3A and 3B show that a recent strong bottleneck occurred approximately 2,000 generations ago (17,500 years), when population size decreased from more than 10,000 to less than 1,000 in about 1,000 generations. The model suggested an ancestral population size of about 15,000 in ancient times, 30,000 generations or 262,000 years ago (the ultimate peak in the TMRCA distribution, Fig. 5). However, other evolutionary schemes could lead to a similar distribution of TMRCA, such as recurrent immigration from a large population into the Moray Firth. Thus, there are several possible interpretations for this pattern. The population may derive from ancestral fragmentation followed by permanent introductions from a metapopulation made of similar colonies, or it may be the result of a sharp decline some time in the past. Several arguments support the bottleneck hypothesis: the gaps in the distribution of repeat numbers Appendix S1C, the rather low immigration rate, and the high probability that coalescent events occurred recently. A more detailed answer will require archaelogical samples (ancient DNA) or samples from a number of distant populations.

During the period corresponding to the drastic bottleneck detected, the current geographical range of this population would have been uninhabitable as it was covered by the last British Ice Sheet (Bradwell et al., 2008), and this ancestral population would likely have re-distributed to the margins of the ice sheet. Whilst the size of large terrestrial mammal populations in Europe is likely to have been reduced during this period (Marshall et al., 1982), harbour seals breed on ice in certain areas, and these colder conditions could have led to population expansion. For example, analysis of mtDNA variation in Antarctic ice-breeding seals indicated that effective population size expanded suddenly during times of intensified glaciations (Curtis, Stewart & Karl, 2009), while southern elephant seals, which require access to open beaches, show the opposite trend (De Bruyn et al., 2009). The strong bottleneck detected at around 17,500 years ago appears to have occurred in the period following the retreat of ice from the current study area, as glacial moraines in the Inverness Firth indicate that the ice margin occurred here around 15,000 years ago (Merritt, Auton & Firth, 1995). Subsequently, the harbour seals may have recolonized this area once the ice retreated, as was the case for ringed seals which colonized the Baltic Sea basin soon after deglaciation (11,500 years) (Palo et al., 2001).

The analysis of TMRCA revealed a peak of coalescence events around 800 generations (7,000 years) ago in a period when mean effective size was approximately 5,000, and may correspond to a transient sharp decrease of effective size. This coincides with the warmest stages in the post-glacial period in the mid-Holocene 8,000-7,000 years ago (Andersson, 1902; Andersson, 1909; Seppä, Bjune AE & Veski, 2009). Current global distribution patterns indicate that harbour seals could have been restricted to more northerly waters at this time. Archaeological records also suggest that populations of harbour seals in the eastern Baltic were founded around 8,000 years ago (Härkönen & Johannesson, 2005), which may also reflect a northward shift in distribution during this period. Here, we see that this period may correspond to a transient decrease of the harbour seal population, long after it was derived from a larger ancestral population. Several dramatic events therefore appear to have influenced the structure and effective size of this population. This complexity inevitably brings uncertainty, particularly as the model cannot fully account for migration events.

Assuming immigration from a large external population, the distribution of coalescence time TMRCA depends on the proportion m of immigrants per generation and on the ratio ε of the considered population size (Ne) to the size of the larger external population (Nikolic & Chevalet, 2014a). Using VarEff provides the distribution of TMRCA which is converted to past effective sizes in an isolated population. In the case of small 4*Ne*m, such as found from the harbour seal analysis, the estimated population size remains in the order of magnitude of the true Ne while increasing with m (Nikolic & Chevalet, 2014a). However, for larger immigration rates from a larger external population, a false current bottleneck could be predicted in a small population. Applying the equation (8) of Nikolic & Chevalet (2014a), with hypothetical immigration (2–4%) from larger populations of harbour seals around the UK (Lonergan et al., 2007, which brings ε about 0.05 to 0.10) and specifically around our North-eastern Scottish population, we estimated the time of a potentially false bottleneck at 100–200 generations ago. This period matches with the last event detected in Fig. 4. Thus, immigration from an external population could potentially lead to a false current bottleneck (100–200 generations ago), and we have to be cautious in the final decline in effective size observed (Fig. 4). We encourage studies on seal migration in this area by the monitoring of individuals between the geographically distinct genetic clusters in UK and neighbouring waters.

Following the same analysis, the older bottlenecks detected at 800 and 2,000 generations ago (Figs. 3 and 5) should not be an effect of immigration. Either way, it is clear that these past events led to a decrease in population size resulting from natural environmental changes. This study suggests that, more recently, the harbour seal population has declined over the last century at a rate of 2–6% in terms of effective size. This downward trend could also result from natural changes in environmental conditions, or from human exploitation that is known to have occurred at many European and Arctic sites over the last millenia (Härkönen & Johannesson, 2005; Murray, 2008).

Conservation implications

Previous population genetic studies of UK harbour seals have focused on using molecular tools to support conservation management by identifying appropriate spatial management units (Olsen et al., 2017). Our study illustrates how genetic tools based on coalescent theory can also provide insights into evolutionary history and temporal changes in population size within those management units. Critically, they contribute knowledge about the state of a population by estimating changes in effective size. Estimates of the current effective population size of this harbour seal population are small compared to theoretical estimates of the minimum viable population size (Reed et al., 2003). Our analyses indicate that the harbour seal population in the north-east UK has remained at a broadly similar level following the bottleneck that occurred after post-glacial recolonization of the area but have slowly declined more recently. Subsequent demographic studies indicate that the Moray Firth population has declined further, by approximately 40%, since the early 1990’s when our DNA samples were collected (Thompson et al., 2019). Thus, whilst contemporary estimates of Ne are expected to remain well above the critical threshold of Ne = 50 suggested to avoid short-term effects of inbreeding (Franklin, 1980) they will now be closer to the Ne = 500 potential threshold for maintainence of long-term adaptive potential (Franklin, 1980). Molecular evidence of connectivity within the much larger European metapopulation exists (Olsen et al., 2017), but further work is required to better understand the extent to which contemporary movements between management units may offset these risks. Contrasting population trends in different UK management units (Thompson et al., 2019) also highlight the need for conservation managers to identify regional demographic drivers. Crucially, potential drivers include both anthropogenic stressors (which might require conservation inerventions) and others such as competition with recovering grey seal populations (Halichoerus grypus) (Wilson & Hammond, 2019); where intervention is less likely to be appropriate. Previous studies in the Baltic Sea indicate that populations of harbour and grey seal have co-varied over historical time-scales (Härkönen & Johannesson, 2005). Our work illustrates how parallel coalescent studies on material from sympatric grey and harbour seal populations in different regions now provides an opportunity to inform conservation practices by exploring potential interactions between these two species at much larger temporal and spatial scales.

Supplemental Information

Supplemental Information 1 Appendices and Supplemental Information

Click here for additional data file.

The authors wish to thank Shaneve Tripp (NYU School of Law) and Wendy West (DAFF) for their english corrections. Ludovic Hoarau (IFREMER) for his help on ArcGis. Katia Feve (INRAE) for her help with the DNA extraction protocol. DNA samples were extracted at INRAE and genotyped at the Toulouse Genopole Platform (http://www.genotoul.fr/). Anonymous reviewers provided many helpful comments on an earlier version of the manuscript.

Additional Information and Declarations

Competing Interests

Author Contributions

Animal Ethics

Data Availability

The authors declare there are no competing interests.

Natacha Nikolic conceived and designed the experiments, performed the experiments, analyzed the data, prepared figures and/or tables, authored or reviewed drafts of the paper, and approved the final draft.

Paul Thompson performed the experiments, authored or reviewed drafts of the paper, and approved the final draft.

Mark de Bruyn authored or reviewed drafts of the paper, and approved the final draft.

Matthias Macé and Claude Chevalet performed the experiments, analyzed the data, authored or reviewed drafts of the paper, and approved the final draft.

The following information was supplied relating to ethical approvals (i.e., approving body and any reference numbers):

Blood samples were collected under Home Office licence issued to the University of Aberdeen under the Animal (Scientific Procedures) Act 1986 (PPL number 60/01351).

The following information was supplied regarding data availability:

Data is available at INRAE:

Nikolic, Natacha; Thompson, Paul; De Bruyn, Mark; Macé, Matthias; Chevalet, Claude, 2020, “Microsatellite data from: Evolutionary history of a Scottish harbour seal population”, https://doi.org/10.15454/AOZ7JI, Portail Data INRAE, V2.

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
