# Peer review of "Evolutionary history of a Scottish harbour seal population"

_PeerJ, doi:10.7717/peerj.9167_

## Round 0.1 · original submission · Minor Revisions

Please pay particular attention to Reviewer 3's suggestions for placing this study in context and the suggestions from Reviewers 1 & 2 to clarify a few of the methods and results.

Reviewer 1 ·

Basic reporting

No comment

Experimental design

No comment

Validity of the findings

no comment

Additional comments

This is a scientifically sound manuscript that evaluates the genetic variability and describes the demographic history of harbor seals in Moray Firth, North East Scotland using a good number of nuclear markers (17 microsatellite loci). The data are properly analyzed employing statistical software adequate to the nature of the markers. The study relies on standard techniques for quantification of genetic variation such as observed and expected heterozygosity, Bayesian analysis, estimation of F-statistics, assignment tests, and factorial correspondence analysis. The demographic history evaluation was tackled using two approaches to estimate present and historical effective population size (MSVAR and VAREFF), which are appropriate for this kind of data. The results presented in this article can be of importance for conservation of this species. I only have one minor concern that could be easily solved in the discussion. The samples used in this study were collected almost 30 years ago. Even though, this does not affect the historical population events detected here, the “actual” effective population size is in fact not that recent, therefore, it should be discussed. The recommendation of “accept, with minor revision” refers mostly to minor revisions and some desired clarifications, which are enumerated in the paragraph-itemized comments below.

Abstract
Line 20: “microsatellites markers” should be “microsatellite markers”
Line 22: “North eastern” should be “Northeastern”

Methods
Line 133-134: It is not clear how the markers were tested first and then the primers defined. If the microsatellites were already described (lines 130-133), then the primers were designed too, or where they redesigned and then tested again? Please, explain.
Line 141: Delete space in “MgCl2 ,”
Figure 2: X-axis legend says 1+2. It should be 1-2

Results
Line 395-397: It seems the ancestral effective size estimated with MSVAR is much larger (128,000-340,000) than with VAREFF (65,000), however it says is same order of magnitude. Please clarify.
Line 401: Reference is missing.
Line 433-434: Figure letters should be capitalized (e.g. Figure 3A). Please check throughout the document.
Line 437: The decline starting at around the las 100 generations is clear; however, it is hard to see if the population remained constant after that in figure 4. Please, explain in more detail why it is assumed that the population remained constant.

Figure 4B: the x-axis is confusing. The legend indicates from calendar year 1995 to 500 years ago (where 500 years ago means 1520), however, it seems to be from year 500 to 1995, correct?

Discussion
Line 564: A comma is needed after “However”
Line 611: U.K. should be UK. It has to be consistent throughout the document.

Reviewer 2 ·

Basic reporting

No comment.

Experimental design

No comment.

Validity of the findings

No comment.

Additional comments

This study characterizes the evolutionary history of harbor seals in North-East Scotland. Specifically, the authors assess changes in effective population sizes at one Special Area of Conservation over time using measures of current, neutral genetic differentiation and population structure. The data and study are novel and, in my opinion, of great importance given that they may be used directly for conservation of a species whose populations have undergone drastic declines. For that reason, the study is highly deserving of publication. The authors have done excellent and thorough work, especially with their methods. They give plenty of detail on how the data were collected, how microsatellites were chosen and scored, and how analyses were carried out. The methods are very thorough, using multiple software programs to verify estimates of genetic differentiation and robust models for inferring historic effective population size. They also explain why their chosen method for assessing coalescence (MSVAR) is more favorable than other, similar methods.

My only overall concern is some apparent inconsistencies between the data presented in the tables/figures and the interpretations of those data.

For example: lines 323 – 325 list the loci that significantly deviated from HWE. Table 1 appears to show the P values from the exact tests for each loci. But, that column is described as the “probability of deviations from HWE” in the table caption. I would then read the values in the table as the probability that they deviate from HWE, rather than as p-values indicating significant deviation. I would suggest describing that column simply as “HWE p values” or something like that rather than as “probability of deviations from HWE”.

Additionally, on line 340, the p value for HWE for the whole seal population is given as P = 0.519, but I do not see this value in Table 1. Please clarify this.

Lines 409 – 421: I found the results from VAREFF on effective population sizes to be confusing and inconsistent at times. For example, lines 415 – 416 indicate that the different measures of effective population size through time (arithmetic mean, harmonic mean, mode, median) were “similar”. This is not consistent with what is shown in Figure 3. In Figure 3A, estimates of mode (in blue) and median (in black) look similar, but the arithmetic mean and harmonic mean look drastically different in effective population size over time. The word “similar” should be clarified and explained in further detail here.

Other concerns:

For analyses in STRUCTURE (in the Clustering analysis section of the methods, lines 216 – 225), why were both STRUCTURE HARVESTER AND Structure Selector used for structure inference? I would suggest only using one of these programs to reduce confusion and wordiness. Additionally, which structure inference methods were used to estimate the best K. Both Maximum Likelihood (Evanno et al 2005) and Puechmaille (2016) methods are mentioned, but it is not currently clear which method was used (or if both were used).

Additionally, lines 363 – 366 describe STRUCTURE and DAPC results, but no data from these analyses are shown, either in tables or in figure format. For greater accessibility to the reader, I would suggest showing at least the Delta K (or Puechmaille) results in a supplementary table and the DAPC results in a figure. I think that would give greater clarity to the purpose behind those analyses.

Lines 525 – 527 give a definition of effective population size according to Frankham 1995, but this is a highly simplified definition, since we know that effective population size is directly related to genetic drift. Since the paper is focused largely on estimating effective population size, I would like to see a more detailed and thorough explanation of the genetic consequences of low effective population size.

I would like to see one final paragraph in the conclusion discussing the potential consequences of continually decreasing effective population size in this seal population. In addition, I would like to see a bit more discussion of how these results, if at all or if possible, will contribute to conservation efforts for these seals or for other species.

Minor considerations:

Line 20: “microsatellites” should be made singular to “microsatellite”

Line 46: omit the word “the” before “abundance”

Line 72: “is” should be made past tense to be consistent with the rest of the methods

Line 108: Omit the word “have” and change “an” to “a” before “SAC”

Line 155: “seal” should be plural as “seals”

Line 228: “Performed” should be lower case

Line 236: There should be an “a” before “Mantel Test”

Line 288: Should “Time to the Most Recent Common Allele” be “Time to the Most Recent Common Acestor” instead?

Line 317: The number of decimal places given for heterozygosity here in the text is two (0.38), but in Table 1, this value is given as 0.384, if I am reading correctly. The number of decimal places should be kept consistent between the text and tables.

Line 359: at the end of this line, I believe “no” should be “not”

Line 363: at the end of this line, I believe the word “support” should be “supported”

Line 366: at the beginning of this line, I believe “is” should be “was”

Line 509: “Association” should be “associated”

Line 522: At the end of the line, the word “thus” should be moved before the word “does”

Line 621: The word “leading” should be “led”

The figures do not appear to have all been created using the same software, so they are not formatted uniformly. Please consider using a single medium, ie. R or Excel, for all figures.

Figure 2: the label for the second tick mark on the x-axis should be 1-2

Figure 3 B,C: Please increase font size on y-axis.

Figure 5: Please increase font size

Reviewer 3 ·

Basic reporting

The majority of the manuscript reads very well with clear, professional and technical english. There are a handful of places where I have made some edits and corrected spelling and grammatical typos. Mostly, the manuscript is well supported by the relevant literature. There are some papers I have suggested throughout to include as they may support some of the authors statements. There are also some noted referencing errors in the text: ommisions of “et al.” in a handful of locations, incorrect years (eg. Hanson et al. should be 2017 and not 2015) and in-text citations that do not appear in the reference list (eg. Hall et al. In press and Thompson et al. In press). There is an absence of the most up-to-date counts and trends in harbour seals (which are available in Thompson et al. 2019), with references pertaining to the time period when the samples were collected (30+ years ago). Although these references are appropriate for this manuscript, I also think it is important to include the most updated data.
The manuscript is well laid out with professional structure, figures and tables. Some notes have been made in the figures/tables to help clarify them for the readers. This manuscript is a self-contained study with a direct connection between the hypotheses, methods and findings. Appropriate scope is provided from outside studies.

Experimental design

Although an extension to other harbour seal studies in the UK, this study has unique aims that fall within the scope of this journal. Both biological and environmental studies are covered in this study. The authors clearly state the research question of the sutdy in final paragraph of the introduction. They are also clear about the current state of knowledge regarding harbour seal biology as well as population structure. It is made clear that the study is acknowledging the knowledge gap regarding historic estimates of population demography and changes. The sample size seems adequate considering the population size of the region and the difficulties of obtaining samples from seals. Although a slighly older technology, microsatellites are still perfectly suited for the type of analyses and questions approached in this study. Ethical considerations are clearly stated with appropriate permits. There is insufficient detail in the lab portion of the methods section to replicate the study. I have left comments in the text with regards to the DNA extraction and PCR steps where some extra detail should be included.

Validity of the findings

The study design, results and their interpretation are valid and robust. The findings contribute nicely to the growing field of conservation genetics and fill a knowledge gap referring to the demographic hisotory of a protected population of harbour seals. With the suggested additions to the laboratory section, the methods should be reproducible for other researchers to apply in other wild populations. By providing the sample metadata and microsatellite genotypes, the data can be re-analysed following the computer-based methods.

Additional comments

a. Due to the nature of working with genetics and marine mammals specifically, I understand that sometimes studies are conducted because the data is available and not always as a result of a driving question. Although impact and novely are not assessed in this review, the manuscript would benefit from more information as to what it is about the Moray Firth specifically (compared to the other UK Harbour seal populations) that warrants a more in-depth analysis than the structure and diversity results that have been found in previous studies (such as Oslen et al. 2017) with added microsatellite loci? Was this location more stable during the two PDV outbreaks? What is the current population trend (since this paper focuses exclusively on the population approximately 30 years ago when the samples were obtained)?
b. Estimating historic abundance, effective population sizes and bottlenecks are very interesting and important for understanding the current status of a population. Although the manuscript mentions that harbour seals are protected, it would be helpful for the reader to understand just how impacted UK harbour seal populations have been in recent decades (using total declines or percentages, see Thomspon et al. 2019: “The status of harbour seals (Phoca vitulina) in the UK”, Aquatic Conservation: Marine and Freshwater Ecosystems, 29:S1).
c. The Moray Firth harbour seals (Loch Fleet) are one of the best studied populations of harbour seals providing the only estimates of vital rates (birth rates, survival rates, fecundity, etc.) in the UK. I imagine this was a factor in choosing this location to investigate further and if so, I recommend highlighting this invaluable dataset in the manuscript.
d. As one of the justifications for this study seems to be a lack of data for conservation measures, how can your new findings about the historic state of this population be incorporated into contemporary management?
e. The most recent estimates of harbour seal abundance in the UK are now available in Thompson et al. 2019. It was reported that mean counts decreased by ~40% between the 1995 and 2005-2016 surveys down to about 900 animals. Although your data come from animals sampled before these more recent declines, it would be advisable to incorporate the more recent into your conclusions and justificaitons for understanding the historical abundances.

Annotated reviews are not available for download in order to protect the identity of reviewers who chose to remain anonymous.

---

## Round 0.2 · accepted · Accept

Thanks for sending this story to PeerJ and for doing such a thorough job of revising your manuscript.